# Comparison of Characteristics of a ZnO Gas Sensor Using a Low-Dimensional Carbon Allotrope

**DOI:** 10.3390/s23010052

**Published:** 2022-12-21

**Authors:** Jihoon Lee, Jaebum Park, Jeung-Soo Huh

**Affiliations:** 1Department of Convergence and Fusion System Engineering, Institute of Global Climate Change and Energy, Kyungpook National University, Daegu 41566, Republic of Korea; 2Department of Energy Convergence and Climate Change, Kyungpook National University, Daegu 41566, Republic of Korea

**Keywords:** gas sensor, ZnO, graphene, carbonnanotube, formaldehyde

## Abstract

Owing to the increasing construction of new buildings, the increase in the emission of formaldehyde and volatile organic compounds, which are emitted as indoor air pollutants, is causing adverse effects on the human body, including life-threatening diseases such as cancer. A gas sensor was fabricated and used to measure and monitor this phenomenon. An alumina substrate with Au, Pt, and Zn layers formed on the electrode was used for the gas sensor fabrication, which was then classified into two types, A and B, representing the graphene spin coating before and after the heat treatment, respectively. Ultrasonication was performed in a 0.01 M aqueous solution, and the variation in the sensing accuracy of the target gas with the operating temperature and conditions was investigated. As a result, compared to the ZnO sensor showing excellent sensing characteristics at 350 °C, it exhibited excellent sensing characteristics even at a low temperature of 150 °C, 200 °C, and 250 °C.

## 1. Introduction

Owing to significant economic growth and urban density, many buildings and facilities are aging over time. To fulfill the changing tastes and increasing needs of people for the amenities with changing times, the reconstruction and construction of new buildings is underway in many localities [1]. In addition, owing to the changing trends, various complex phenomena are occurring, such as construction and living in ordinary houses and densely populated areas. Accordingly, various compounds volatilized from chemicals, such as preservatives, adhesives, and paints, used inside and outside the newly built buildings contaminate the indoor and outdoor air, and these contaminants directly or indirectly affect the human body and cause harmful reactions [2,3,4,5].

Among the various compounds that pollute the indoor and outdoor air and affect the human body, volatile organic compounds (VOCs) are the compounds that are emitted the most within six months of a building’s construction, and most of them are classified as carcinogens [6,7,8]. Among these carcinogens, formaldehyde is a representative carcinogen that is easily volatilized into the atmosphere and has harmful effects on the human body [9,10,11]. The formaldehyde poisoning symptoms differ from person to person. In addition, the concentration ranges of the pollutant compounds sufficient to affect the body may show remarkable differences. The American Conference of Governmental Industrial Hygienists, for example, reported that an exposure to more than 0.3 ppm of formaldehyde over a short period of 15 min can be dangerous to one’s health. Because various compounds are mixed, an individual compound cannot distinguish and confirm their presence, such as smell. Therefore, a gas sensor capable of detecting harmful gases is needed to determine the cause of the hazardous reactions. Harmful gases can be detected by several types of gas sensors, such as semiconductor-, electrochemical-, infrared-, and catalytic reaction-type sensors. Different types of gas sensors present different characteristics and can be used in various environments depending on the application requirements. The most well-known type of gas sensor is the semiconductor-type gas sensor. The detection principle of this type of gas sensor is based on the interaction of the semiconductor surface with the gas analyte; its surface is heated in the atmosphere with a heater. Compared with other types of sensors, semiconductor-type sensors are smaller in size but have a higher sensitivity.

Among semiconductor gas sensors, the ZnO semiconductor gas sensor has a high sensitivity and stability at an about 350 °C operating temperature and is suitable for miniaturization and low-concentration combustible gas detection [12,13,14,15,16,17,18,19,20]. In addition, low-dimensional carbon allotrope, with an excellent electrical conductivity, was coated for the purpose of uniform connection between ZnO nanorods and low-dimensional carbon allotrope through spin coating that can be homogeneously coated [21,22,23].

This study aimed to improve the characteristics of the ZnO-modified gas sensor.

## 2. Materials and Methods

The configurations of the sensor are shown in Figure 1. The sensors have dimensions of 4.5 × 3.78 × 0.3 mm and are composed of a Au electrode and an Alumina substrate. The resistance of the rear Pt heater was ~13 Ω. Thus, the resistance can be measured while controlling the temperature of the sensor surface. The wire connection between the heater and electrode is configured appropriately to prevent a short circuit (Figure 1). The wire used to supply the current to the heater and sensor is made of platinum and has a diameter of 0.05 mm, and gold paste is used for bonding.

Figure 2 shows the manufacturing process for the type A and type B sensors. Figure 3 presented a schematic diagram of the completed sensor following the progression of Figure 2. Pt was deposited on the alumina substrate at 20 mA for 60 s, after which a Zn layer of 1000 Å was deposited using the sputtering equipment. Subsequently, annealing was performed at 600 °C for 1 h to form ZnO seeds in the Zn layer. The sensor systems were divided into A-type and B-type sensors by two processing methods.

The type A sensor was subjected to ultrasonic synthesis after spin coating at 1000 rpm twice with a 0.1 mL graphene and CNT (carbon nano tube) mixture of a 0.1 mg/mL concentration. To stabilize the nanostructures formed after the ultrasonic synthesis, a heat treatment was performed at 400 °C for 2 h.

For the type B sensor, ultrasonic synthesis was employed, and the heat treatment was performed at 400 °C for 2 h to stabilize the nanostructure which was formed. Spin coating was performed twice at 1000 rpm using 0.1 mL of graphene and a CNT mixture solution at a 0.1 mg/mL concentration, and the heat treatment was performed at 100 °C for 10 min.

A 750 W ultrasonic synthesis equipment (Sonic and Materials, Inc.) was used. The tip size of the ultrasonic device was ½ an inch. An ultrasonic frequency of 20 kHz and 300,000 J of energy was applied, and the solutions used to grow the Zn nanostructures were distilled water-based zinc nitrate hydrate [Zn(NO_3_)_2_.6H_2_O] and hexamethylene tetramine [C_6_H_12_N_4_] aqueous solution (0.01 M, 200 mL).

The effect of the addition of graphene and CNTs to the ZnO nanostructure was examined by dividing the sensors into types A and B. The differences in the sensor properties, such as the selectivity and sensitivity, according to the differences in bonding were confirmed. Thereafter, the optimal sensor for the given operating conditions was identified.

Figure 4 shows the process of evaluating the sensitivity and recovery characteristics of the sensor for the measurement of the target gas at ppb levels. The target gas was injected for 5 min after stabilization for 1 h in the air, and the recovery process was continuously performed for 5 min to evaluate the recovery characteristics.

## 3. Results and Discussion

### 3.1. FE-SEM (Field Emission Scanning Electron Microscope)

Figure 5 shows the FE-SEM microstructures of the surfaces of type A and type B samples.

The observed surface was subjected to component analysis through FE-SEM EDS, and through this, CNT and ZnO could be distinguished [24,25]. At this time, ZnO and CNT could be observed in both type A and type B. As a result of FE_SEM, needle-shaped nanorods were observed in both type A and type B, whereas graphene and CNT overlap with ZnO nanorods in type B samples.

### 3.2. XRD (X-ray Diffraction) Analysis

The components and structures of the ZnO sensors prepared using sonochemical synthesis and carbon allotrope spin coating were analyzed using XRD. Figure 6 shows the XRD patterns obtained for both types of samples. The growth directions observed in the pattern are (1 0 0), (0 0 2), (1 0 1), (1 0 2), (1 1 0), (1 0 3), (1 1 2), (2 0 0), and (2 0 2). The presence of crystalline ZnO is confirmed, which is expected to affect the sensitivity characteristics [26].

### 3.3. Raman Analysis

The components and structures of the sensors were analyzed using Raman spectroscopy. The peaks at 437, 520, and 570 cm^−1^ confirm the existence of ZnO [27].

In the Raman analysis, the crystallinity of CNT and graphene can be evaluated using the D and G band intensity ratio, I_d_/I_g_. The lower the ratio, the higher the crystallinity, owing to the increase in the amount of non-crystallized carbon.

As shown in Figure 7 and Figure 8, the value of I_d_/I_g_ for type A is 0.70, which is lower than that for the type B value 1.07.

### 3.4. Comparison of Sensitivity and Recovery for Each Sensor for 1 ppb Level of Formaldehyde

To evaluate the sensitivity and recoverability of formaldehyde, the measurements were performed for 100, 200, 500, and 1000 ppb levels.

The measurement results for the types A and B are shown in Figure 9 and Figure 10, respectively. The sensitivity is represented by S, R_air_ is the resistance value when air is injected, and R_gas_ is represented by the resistance value when a specific gas is injected. The sensitivity is obtained using the following formula:S (%) = ((R_gas_ − R_air_)/(R_air_)) × 100(1)

The target gas was detected by both types of sensors from the 100 ppb level of formaldehyde. The similarity between the two sensors was that the change in the resistance change in the resistor increased as the concentration of formaldehyde increased. The highest sensitivity was exhibited by type A at 200 ℃, and the highest recovery power was exhibited by type B at 200 and 250 ℃. The combination of a good sensitivity and recovery was found to be at 200 ℃ for type B. At 150 °C, both the sensors exhibited a low sensitivity and an unstable graph for their recovery and sensitivity. In terms of the sensitivity toward formaldehyde, 200 ℃ was found to be a suitable operating temperature for both types, A and B. In terms of the recovery, type B showed a good performance at 200 and 250 ℃ for low ppb concentrations and a good performance at 250 ℃ at high ppb concentrations.

In order to compare that the formaldehyde sensing characteristics of the ZnO sensor before coating the carbon allotrope, the ZnO gas sensor was measured and compared based on 200 ℃, which showed good characteristics in type A and type B, shown in Figure 11. When 100ppb of formaldehyde is injected, the sensitivity appears to change, but it recovers differently from the low-dimensional carbon allotrope sensor in the recovery.

### 3.5. Comparison of Sensitivity and Recovery for Each Sensor for 1 ppb Level of Toluene

To evaluate the sensitivity and recoverability of toluene, the measurements were performed for 100, 200, 500, and 1000 ppb levels in Figure 12 and Figure 13.

The target gas was detected by both types of sensors from the 100 ppb level of toluene. The similarity between the two sensors is that the change in the sensitivity increased as the concentration of the toluene increased. Type A showed the highest sensitivity at 200 °C, and type B showed the highest recovery at 200 °C. However, unlike formaldehyde, an immediate reaction did not appear. Additionally, the sensitivity changed indiscriminately. During recovery, it shows a relatively slow change with formaldehyde, and even after a certain recovery progresses, an unstable sensitivity appears. Through this, the difference between formaldehyde and toluene can be confirmed.

## 4. Conclusions

Herein, we investigated the performance of a low-dimensional carbon allotrope and ZnO gas sensor for the measurement of formaldehyde and toluene gas. Compared to the pure ZnO sensor, which showed excellent sensing characteristics only at a high temperature of 350 °C, excellent sensing characteristics were obtained even at a low operating temperature of 200 °C when ZnO was combined with a low-dimensional carbon allotrope [28]. The type A sensor showed a high sensitivity of about 60% at 100 ppb of formaldehyde and the type B sensor showed a sensitivity of about 50% at 100 ppb of formaldehyde, and type B confirmed that the recovered value returned to the value before the gas measurement. Accordingly, type A can be used in situations that require a rapid measurement and response, and type B can be used for the long-term measurement and can continuously and consistently observe high concentrations of formaldehyde.

Additionally, in toluene, it was confirmed that the best properties appeared at 200 ℃, the same as formaldehyde. However, it shows a different tendency than formaldehyde, so it can differentiate between toluene and formaldehyde.

We demonstrated a method that can incorporate low-dimensional carbon allotropes in the process of forming ZnO nanostructures rather than through an individual fabrication sensor and obtained optimal sensing characteristics for the combined graphene and CNT sensors. Through subsequent research, it will be possible to develop a graphene and CNT combined sensor that has the advantages of type A and type B.

## Figures and Tables

**Figure 1 sensors-23-00052-f001:**
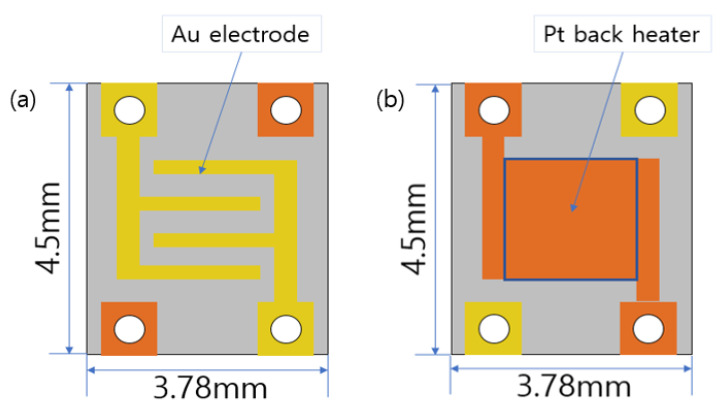
Configuration diagram of the sensor substrate (**a**) front side (**b**) rear side.

**Figure 2 sensors-23-00052-f002:**
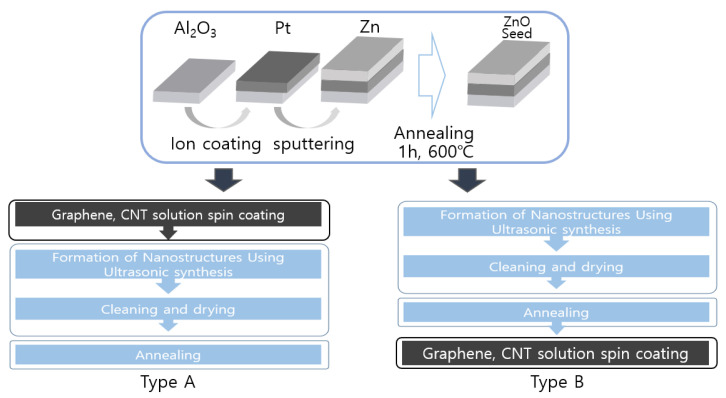
Manufacturing process for the type A and type B sensors.

**Figure 3 sensors-23-00052-f003:**
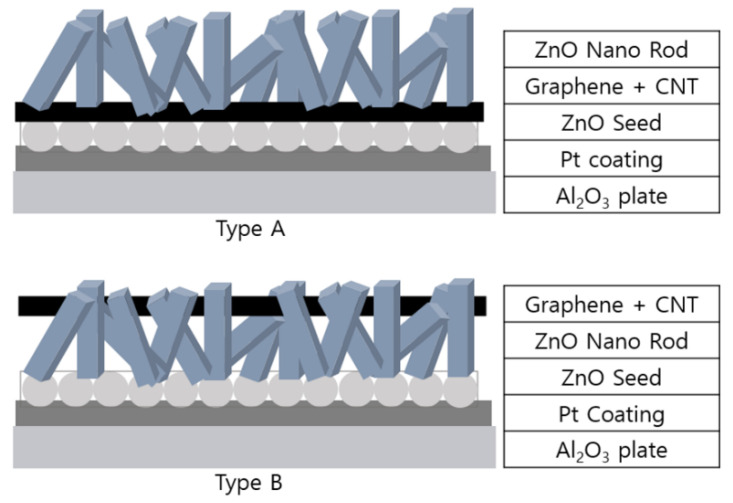
Schematic of the type A and type B sensors.

**Figure 4 sensors-23-00052-f004:**
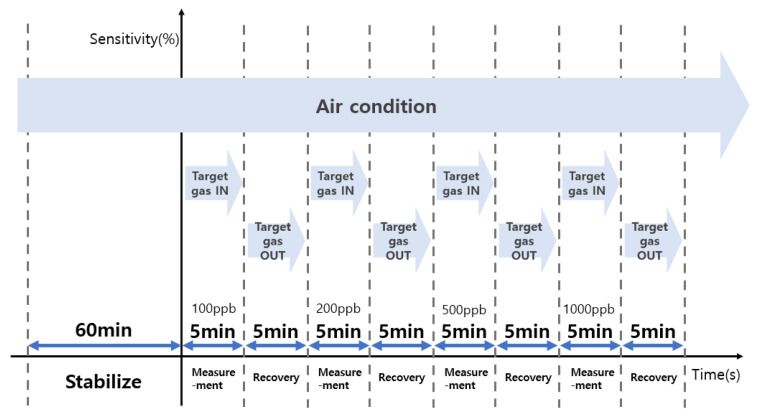
Process for determining the sensing characteristics at ppb levels of the target gas.

**Figure 5 sensors-23-00052-f005:**
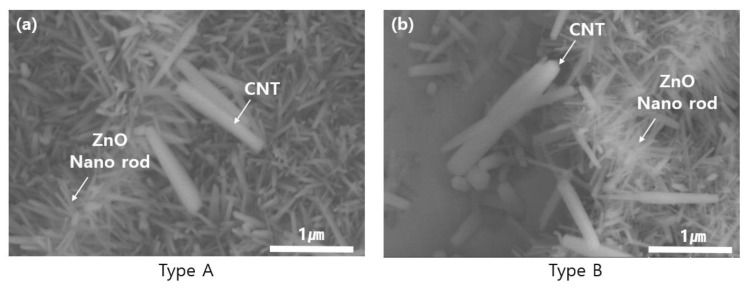
FE-SEM image of samples (**a**) type A (**b**) type B.

**Figure 6 sensors-23-00052-f006:**
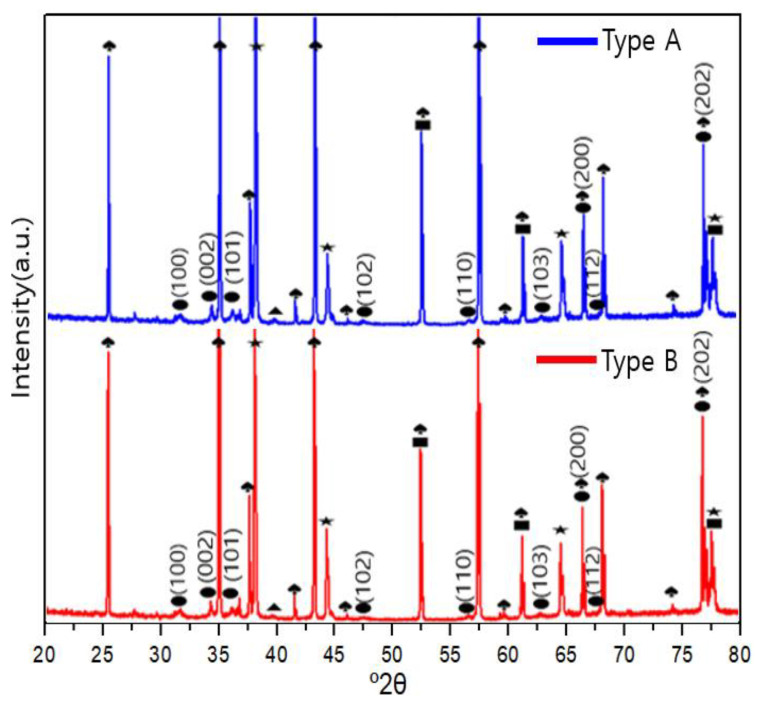
XRD patterns for the type A and type B samples. Patterns show Al_2_O_3_ (♠), Au (★), ZnO (●), C (■), and Pt (▲), and the components indicated by each peak are shown.

**Figure 7 sensors-23-00052-f007:**
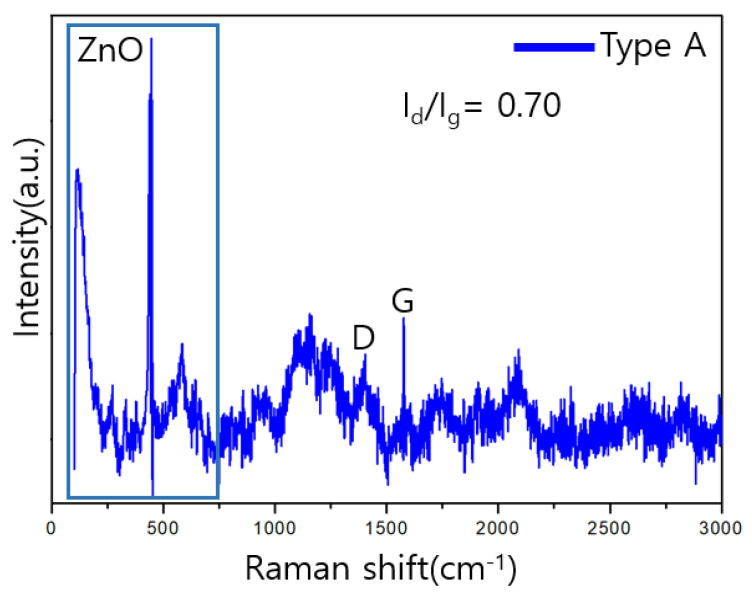
Raman analysis results for type A samples.

**Figure 8 sensors-23-00052-f008:**
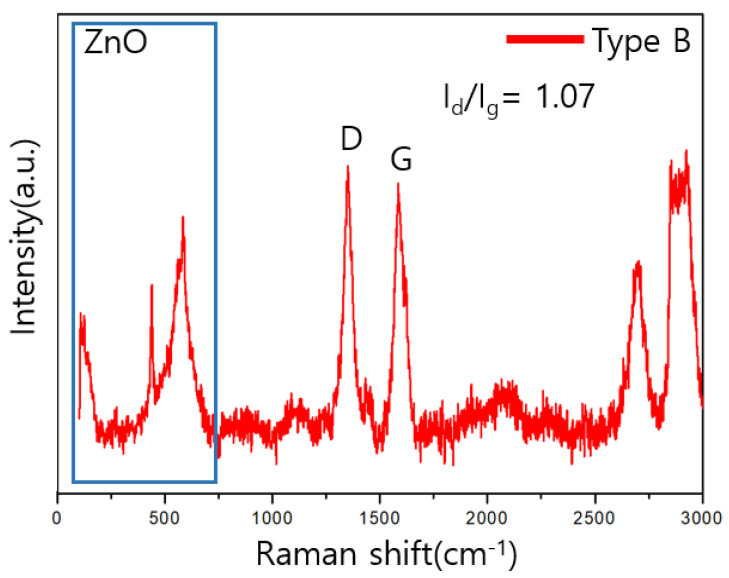
Raman analysis results for type B samples.

**Figure 9 sensors-23-00052-f009:**
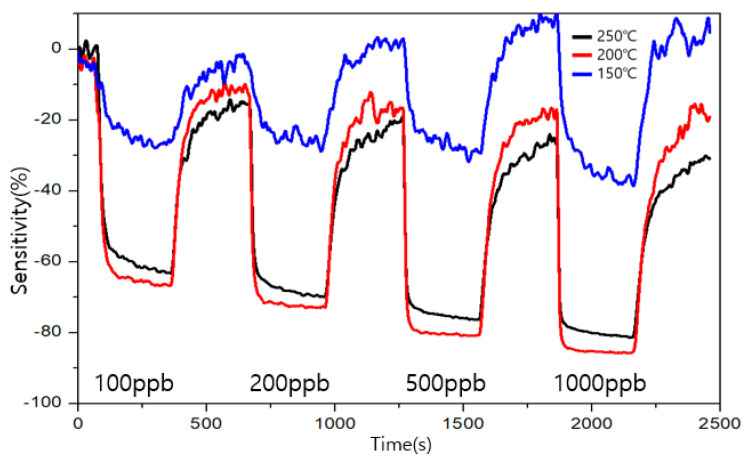
Sensitivity and recovery at different operating temperatures for formaldehyde in the ppb range for the type A sensor.

**Figure 10 sensors-23-00052-f010:**
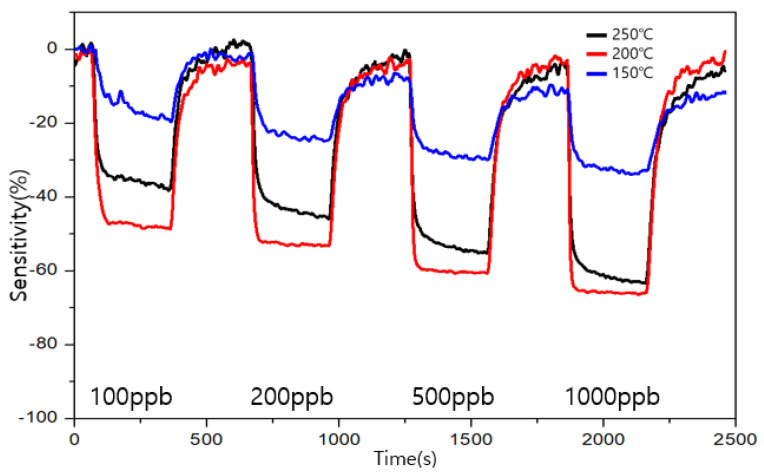
Sensitivity and recovery at different operating temperatures for formaldehyde in the ppb range for the type B sensor.

**Figure 11 sensors-23-00052-f011:**
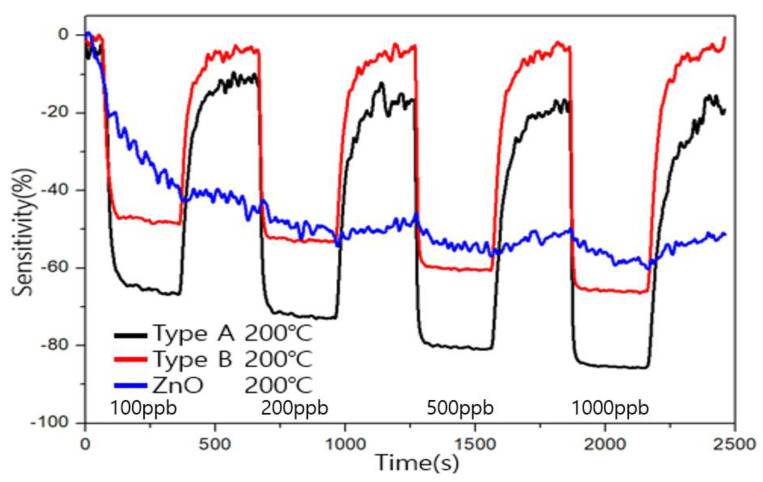
Comparison of sensitivity and recovery to formaldehyde at 200 °C between low-dimensional allotrope modified ZnO sensor (type A and type B) and ZnO sensor.

**Figure 12 sensors-23-00052-f012:**
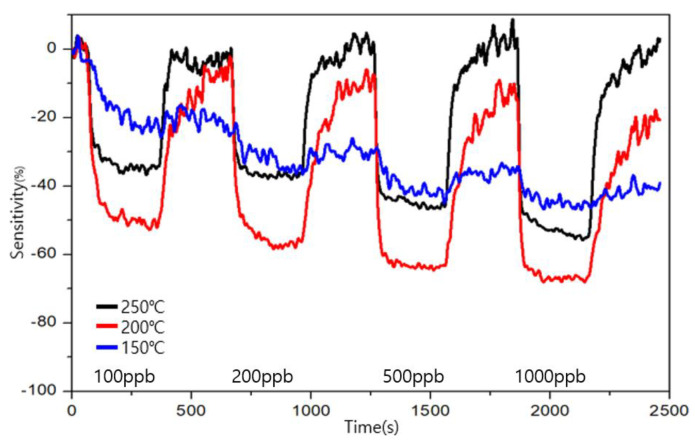
Sensitivity and recovery at different operating temperatures for toluene in the ppb range for the type A sensor.

**Figure 13 sensors-23-00052-f013:**
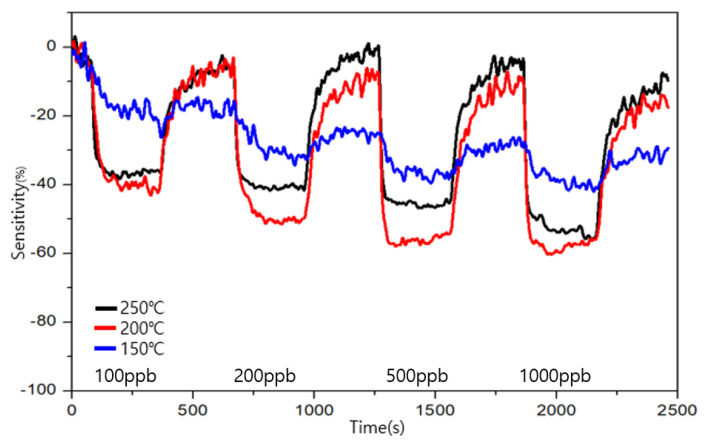
Sensitivity and recovery at different operating temperatures for toluene in the ppb range for the type B sensor.

## Data Availability

The data that support the findings of this study are available from the corresponding authors upon request.

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
