# Peer review of "Comparison of Characteristics of a ZnO Gas Sensor Using a Low-Dimensional Carbon Allotrope"

_sensors, 2022, doi:10.3390/s23010052_

Round 1

Reviewer 1 Report

Jihoon Lee et al., presented a work “Comparison of Characteristics of a ZnO Gas Sensor using a 2 Low-dimensional Carbon Allotrope.” Paper was well written. However, before being accepted, authors should answer to the following comments.

Comments

1. Describe the list of chemicals which were used in this work .

2.What is the detection limit of this work. How do you calculate detection limit from your experimental results.

3. You must describe the XRD results of synthesis first in characterization section.

4. “The peaks at 437, 520, and 570 cm-1 confirm the existence of ZnO” (Line no. 138), provide the evidence with reference.

5 English language needs to be improved.

Author Response

Response to Reviewer 1 Comments

Thank you for your thorough review and valuable comments. We made point-to-point reply to the reviewer’s comments and revisions are reflected in the revised manuscript marked in red.

Point 1: Describe the list of chemicals which were used in this work.

Response 1: We mentioned the list of chemicals in paragraph 2. Materials and Methods. (line 105-107)

Point 2: What is the detection limit of this work. How do you calculate detection limit from your experimental results.

Response 2: The detection limit is dependent on value of â–³R=Rgas – Rair. Sensitivity is measured in the S (%) = ((Rgas - Rair)/(Rair))*100. We mentioned in paragraph 3.4. (line 166)

Point 3: You must describe the XRD results of synthesis first in characterization section.

Response 3: XRD results were shown in Figure 8 after synthesis of type A and type B

Point 4: “The peaks at 437, 520, and 570 cm-1 confirm the existence of ZnO” (Line no. 138), provide the evidence with reference.

Response 4: According to your comment, the existence of ZnO peaks in Raman spectrum were added in paragraph 3.3 which is referred with reference 24.

Point 5: English language needs to be improved.

Response 5: We received an English language review by a native English speaker through Editage editing institute and referred to this information in the ‘Acknowledgements’.

Reviewer 2 Report

For the present revised manuscript (Comparison of Characteristics of a ZnO Gas Sensor using a Low-dimensional Carbon Allotrope), the authors fabricated semiconductor gas sensors using zinc oxide and low-dimensional carbon allotropes.  The sensitivity and response time of two different types of sensors, A and B were studied. I suggest the acceptance to this work for publication with a major revision.

1.     The PDF card number should be given in the XRD.

2.     Figure 7. modify the clarity to improve the readability.

3.     It is recommended that the author integrate the chart appropriately.

4.     The sensor test process should be present in the “2. Materials and Methods”.

5.     The authors should describe the differences in the distribution of nanorods in the two SEM images in detail. From the SEM images it is difficult to distinguish ZnO and CNT nanorods.

6.  Formula 1, Please clearly indicate the physical meaning represented by each symbol and the accurate definition of sensitivity for the convenience of readers.

7.     There are some grammatical and language errors, please check the English grammar of this manuscript carefully. e.g., It is recommended that the font of “seed” in Figure 3 be consistent with ZnO.

Author Response

Response to Reviewer 2 Comments

Thank you for your thorough review and valuable comments. We made point-to-point reply to the reviewer’s comments and revisions are reflected in the revised manuscript marked in red.

Point 1: The PDF card number should be given in the XRD.

Response 1: The PDF card number in XRD is 04-015-8996: Al2O3, 04-001-2616: Au, 01-080-3002: ZnO, 04-015-1012: ZnAl2O4, 00-018-0311: C, 04-016-6405: Pt.

Point 2: Figure 7. modify the clarity to improve the readability.

Response 2: According to your comment, we modified the clarity to improve the redadability in Figure 8.(Figure 7. is changed to Figure 8.)

Point 3: It is recommended that the author integrate the chart appropriately.

Response 3: According to your comment, we modified the chart appropreiately. 

Point 4: The sensor test process should be present in the “2. Materials and Methods”.

Response 4: According to your comment, the sensor test process section was moved to the 2. Materials and Methods (line 112-118, Figure 5.)

Point 5: The authors should describe the differences in the distribution of nanorods in the two SEM images in detail. From the SEM images it is difficult to distinguish ZnO and CNT nanorods.

Response 5: According to your comment, the difference between ZnO and CNT was monitored through EDS of FE-SEM and added in paragraph 3.2 FE-SEM (line 130-135)

Point 6:.Formula 1, Please clearly indicate the physical meaning represented by each symbol and the accurate definition of sensitivity for the convenience of readers.

Response 6: According to your comment, we added the physical meaning represented by each symbol and the accurate definition of sensitivity in paragraph 3.4 Comparison of Sensitivity and Recovery for each Sensor for 1 ppb Level of Formaldehyde (line 163-166)

Point 7:.There are some grammatical and language errors, please check the English grammar of this manuscript carefully. e.g., It is recommended that the font of “seed” in Figure 3 be consistent with ZnO.

Response 7: According to your comment, we modified the English grammar and language errors.

Reviewer 3 Report

Title:

1.            The title is confusing. Do you mean modification of a ZnO Gas Sensor using a Low-dimensional Carbon Allotrope or comparison of … to pure ZnO sensor?

Introduction:

1.            The authors give a detailed introduction of VOCs and their harmful effects. However, this paper focuses on the comparison of ZnO and carbon allotropes sensors. So, there is a lack of the introduction of the current progress of these sensors.

2.            As I know, the performance of ZnO gas sensors can be improved by doping, hybridization, surface modification, and heterostructure. So why do the authors compare carbon allotropes sensors with pure ZnO, not the modified ZnO? Comparison with pure ZnO is weak evidence of improvement.

3.            Line 58: “Other types of sensors”, whare are other sensors? This is more relevant to this paper than the introduction of VOCs.

Materials and Methods

1.            Clear and concise.

Results and Discussion

1.            3.1: Authors state that “similar ZnO nanorods”. However, based on my observation, nanostructures of type A is more uniform than type B.

2.            3.2: symbols in Figure 7 are elongated in horizontal direction.

3.            3.3: line 141 “vi” typo

4.            3.4: this part is lack of selectivity: how about the resistance change upon exposure to H2O vapor, or other gases (NH3, etc)?

5.            It is unclear to me that what is the key materials to report HCHO, ZnO or carbon allotropes? The authors seem to state that carbon allotropes are the key materials.

If so, why do authors spincoat carbon allotropes on ZnO? I would propose any substrates (maybe conductive) would work and can simplify this device.

If not, this device should be called a carbon allotrope modified ZnO sensor.

Conclusion:

1.            Line 179: “pure ZnO sensor, which shows ….” Is this statement based on authors’ observations (please add the results) or references (please add refs)?

2.            Line 183: “60, 50” should add “%” to make it clear.

Author Response

Response to Reviewer 3 Comments

Thank you for your thorough review and valuable comments. We made point-to-point reply to the reviewer’s comments and revisions are reflected in the revised manuscript marked in red.

Point 1:.The title is confusing. Do you mean modification of a ZnO Gas Sensor using a Low-dimensional Carbon Allotrope or comparison of … to pure ZnO sensor?

Response 1: We compared difference of fabrication structures of ZnO modified gas sensor with  low-dimensional carbon allotrope sensor in figure 3 and figure 4.

Point 2:.The authors give a detailed introduction of VOCs and their harmful effects. However, this paper focuses on the comparison of ZnO and carbon allotropes sensors. So, there is a lack of the introduction of the current progress of these sensors.

Response 2: Our manuscript is targeting the sensing characteristic of harmful gases like       formaldehyde by using ZnO-modifed gas sensors. We reorganized 1.Introduction part.

Point 3:.As I know, the performance of ZnO gas sensors can be improved by doping, hybridization, surface modification, and heterostructure. So why do the authors compare carbon allotropes sensors with pure ZnO, not the modified ZnO? Comparison with pure ZnO is weak evidence of improvement.

Response 3: We added fig 13 in order to differentiate ZnO-midified sensor with pure ZnO sensor.   And in order to compare selectivity between formaldehyde and other VOC gases, we added fig 14 and fig 15 in ZnO-modified sensor with carbon allotrope modification

Point 4: Line 58: “Other types of sensors”, whare are other sensors? This is more relevant to this paper than the introduction of VOCs.

Response 4: we explanned other types of sensors such as semiconductor-, electrochemical-, infrared-, and catalytic reaction-type sensors. (line 51)

Point 5: Clear and concise.

Response 5: We reorganized the Materials and Methods part and manuscript.

Point 6: 3.1: Authors state that “similar ZnO nanorods”. However, based on my observation, nanostructures of type A is more uniform than type B.

Response 6: yes, the nanostructure of type A more uniform are of type B

Point 7: 3.2: symbols in Figure 7 are elongated in horizontal direction.

Response 7: According to your comment, we changed symbols in Figure 8.(Figure 7. is moved to Figure 8.)

Point 8: 3.3: line 141 “vi” typo

Response 8: According to your comment, we changed “vi”.

Point 9: 3.4: this part is lack of selectivity: how about the resistance change upon exposure to H2O vapor, or other gases (NH3, etc)?

Response 9:The sensitivity of H2O vapor and other gas(NH3 etc) is negligible in ZnO modified gas sensors.

Point 10: It is unclear to me that what is the key materials to report HCHO, ZnO or carbon allotropes? The authors seem to state that carbon allotropes are the key materials. If so, why do authors spincoat carbon allotropes on ZnO? I would propose any substrates (maybe conductive) would work and can simplify this device. If not, this device should be called a carbon allotrope modified ZnO sensor.

Response 10: Yes. Our sensor is a carbon allotrope modified ZnO gas sensor (line 61-64)

Point 11: Line 179: “pure ZnO sensor, which shows ….” Is this statement based on authors’ observations (please add the results) or references (please add refs)?

Response 11: According to your comment, we added figure 13 to compare with putr ZnO gas sensor (line 186-193)  

Point 12: Line 183: “60, 50” should add “%” to make it clear.

Response 12: According to your comment, we added to “%” to make it clear.

Round 2

Reviewer 1 Report

Manuscript can be accepted in the present form.

Author Response

Response to Reviewer 1 Comments

comments

: Manuscript can be accepted in the present form.

Response

: We appreciate your recommendations.

Reviewer 2 Report

The current status of the article is not suitable for publication.

The reasons are as follows.

Point 3: It is recommended that the author integrate the chart appropriately.

Response 3: According to your comment, we modified the chart appropreiately.

However, no relevant modifications were found in the text. In fact, the readability of the article is greatly reduced because the author's chart settings are too arbitrary. The content in Figure1and 2 largely overlaps. The SEM images can be integrated in one for comparison.

Point 5: The authors should describe the differences in the distribution of nanorods in the two SEM images in detail. From the SEM images it is difficult to distinguish ZnO and CNT nanorods.

Response 5: According to your comment, the difference between ZnO and CNT was monitored through EDS of FE-SEM and added in paragraph 3.2 FE-SEM (line 130-135)

The EDS characterization data the author mentioned in the response cannot find in the revised manuscript as well. The only thing we can see is the author's groundless discussion “The observed surface was subjected to component analysis through FE-SEM EDS, and through this, CNT and ZnO could be distinguished. At this time, ZnO and CNT could be observed in both Type A and Type B. As a result of FE_SEM, needle-shaped nanorods were observed in both type A and type B, whereas graphene and CNT overlap with ZnO nanorods in type B samples.”

In addition, in combination with Figure 4 given by the author, for the two kinds of sensors, the SEM images should show different morphologies. In other words, the top layer of type A sensor is ZnO, and the top layer of type B sensor is GO and CNT, so it should be distinguished under the SEM. But unfortunately, it is difficult to see the difference in the SEM images Figure 6 and Figure 7 given by the author.

Author Response

Response to Reviewer 2 Comments

comments :

The current status of the article is not suitable for publication.

The reasons are as follows.

Point 3: It is recommended that the author integrate the chart appropriately.

Response 3: According to your comment, we modified the chart appropreiately.

However, no relevant modifications were found in the text. In fact, the readability of the article is greatly reduced because the author's chart settings are too arbitrary. The content in Figure1and 2 largely overlaps. The SEM images can be integrated in one for comparison.

Response

: According to your recommendation of integrating the chart appropriately, we merged the Figures (Figure 1 and Figure 2 revised into Figure 1 and Figure 6 and Figure 7 merged into Figure 5)

Point 5: The authors should describe the differences in the distribution of nanorods in the two SEM images in detail. From the SEM images it is difficult to distinguish ZnO and CNT nanorods.

Response 5: According to your comment, the difference between ZnO and CNT was monitored through EDS of FE-SEM and added in paragraph 3.2 FE-SEM (line 130-135)

The EDS characterization data the author mentioned in the response cannot find in the revised manuscript as well. The only thing we can see is the author's groundless discussion “The observed surface was subjected to component analysis through FE-SEM EDS, and through this, CNT and ZnO could be distinguished. At this time, ZnO and CNT could be observed in both Type A and Type B. As a result of FE_SEM, needle-shaped nanorods were observed in both type A and type B, whereas graphene and CNT overlap with ZnO nanorods in type B samples.”

In addition, in combination with Figure 4 given by the author, for the two kinds of sensors, the SEM images should show different morphologies. In other words, the top layer of type A sensor is ZnO, and the top layer of type B sensor is GO and CNT, so it should be distinguished under the SEM. But unfortunately, it is difficult to see the difference in the SEM images Figure 6 and Figure 7 given by the author.

Response

: In Figure 5, we merged from Figure 6 and Figure 7 with the measurement of FE-SEM. CNT morphology appeared on the ZnO nano-rod structure. And 1~5 % modification of carbon allotropes into ZnO substrates were shown similar morphology and component structures in Figure 5 and Figure 6

Reviewer 3 Report

Thanks for the authors’ effort on the revision of this manuscript.

Point 2 and 3: This request is not fully addressed. Please indicate the advantage of the authors’ ZnO sensor as compared to the existing ones in literature.

Author Response

Response to Reviewer 3 Comments

comments :

Thanks for the authors’ effort on the revision of this manuscript.

Point 2 and 3: This request is not fully addressed.

(Point 2:.The authors give a detailed introduction of VOCs and their harmful effects. However, this paper focuses on the comparison of ZnO and carbon allotropes sensors. So, there is a lack of the introduction of the current progress of these sensors.

Point 3:.As I know, the performance of ZnO gas sensors can be improved by doping, hybridization, surface modification, and heterostructure. So why do the authors compare carbon allotropes sensors with pure ZnO, not the modified ZnO? Comparison with pure ZnO is weak evidence of improvement.)

Response

response point 2 : We compared and improved the sensitivity and selectivity of formaldehyde gas, which is main reason of new house syndrome. ZnO pure sensor is not suitable for this purpose. In order to solve and improve sensing characteristics, we modified ZnO pure sensor with carbon allotropes.

response point 3 : In Figure 11, we compare ZnO sensor with carbon allotrope modified ZnO sensors to see the improvement of sensitivity and recovery to formaldehyde at 200℃ operation temperature. In Figure 12 and 13, we have shown the improvement of sensitivity and recovery for toluene in order to improve the selectivity between formaldehyde and toluene.

Please indicate the advantage of the authors’ ZnO sensor as compared to the existing ones in literature.

Response

: Our ZnO-modified sensor improve the sensitivity and selectivity of formaldehyde sensing characteristics compared to other VOC gases like toluene (line 213-217, line 224-226).
